# Evaluation of Spatial Distribution of Three Major *Leptocorisa* (Hemiptera: Alydidae) Pests Using MaxEnt Model

**DOI:** 10.3390/insects13080750

**Published:** 2022-08-20

**Authors:** Jeong Ho Hwang, Se-Hyun Kim, Sunhee Yoon, Sunghoon Jung, Dong Hee Kim, Wang-Hee Lee

**Affiliations:** 1Natural History Division, National Science Museum, Daejeon 34143, Korea; 2Department of Smart Agriculture Systems, Chungnam National University, Daejeon 34134, Korea; 3Department of Applied Biology, Chungnam National University, Daejeon 34134, Korea; 4Department of Biosystems Machinery Engineering, Chungnam National University, Daejeon 34134, Korea

**Keywords:** Alydidae, rice pest, potential distribution, occurrence possibility, species distribution modeling

## Abstract

**Simple Summary:**

*Leptocorisa* (Hemiptera: Alydidae) species, which are pests to rice cultivation, have changed their habitats owing to climate change, suggesting the precautionary evaluation of the distribution of these pests. In this study, we targeted three major pests in *Leptocorisa* species (*L. chinensis*, *L. acuta*, and *L. oratoria*) and evaluated their potential distributions. Most of the recorded regions of the *Leptocorisa* species are consistent with the result of potential distributions predicted, and the model predicted that climate change would expand their habitats mostly under the effect of minimum temperature and precipitation varied by seasons. The potential distributions of three species were expected to cover major rice cultivation areas regardless of climate change, suggesting a necessity for establishing a sustainable control strategy for the pests.

**Abstract:**

We targeted three major *Leptocorisa* species (*L. chinensis*, *L. acuta*, and *L. oratoria*) and evaluated their potential distributions using MaxEnt. The results showed that most Asian countries and northern Australia would be suitable for at least one of these pest species, and climate change will expand their habitat northward. All of the developed models were evaluated to be excellent with AUC, TSS, and OR10%. Most of the recorded regions of the *Leptocorisa* species are consistent with the result of potential distributions predicted in this study. The results confirmed that the minimum temperature of the coldest month mainly influences the three *Leptocorisa* species distributions. The potential distributions of the three species cover major rice cultivation areas regardless of climate change, suggesting that it would be necessary to establish a sustainable control strategy for the pests.

## 1. Introduction

Rice (*Oryza sativa*) is a major food crop that is produced and consumed by approximately 90% of the population in Asia [1,2]. It is also produced in Oceania, mainly in Australia [3]. Pecky damage, characterized by roughly circular lesions, is an important quality factor in rice because the grains damaged by it weigh less than regular grains, significantly reducing the amount of production as well as the quality [4]. *Leptocorisa* (Hemiptera: Alydidae) species are notorious pests that cause pecky rice, and they are distributed in main rice cultivation areas, suggesting a high risk of damage. *Leptocorisa* species negatively affect rice yield, grain quality, and seed viability, causing severe damage to rice production in various Asian and Oceanian countries [5,6,7,8,9,10,11,12]. Among *Leptocorisa* species, *L**eptocorisa chinensis* Dallas, 1852, *L**eptocorisa acuta* (Thunberg, 1783), and *L**eptocorisa oratoria* (Fabricius, 1764) (syn. *L. oratorius* (Fabricius, 1764)) are the major pests that damage rice in warm temperate, subtropical, and tropical regions in Asia and Oceania [13,14,15]. *L**eptocorisa chinensis* is prevalent in warm temperate climates, whereas *L. acuta* and *L. oratoria* are distributed in subtropical and tropical climates [15,16]. *L**eptocorisa acuta* is more prominent in dry mountainous regions, whereas *L. oratoria* is dominant in the lowlands of tropical regions [15,17].

To control these damages in rice, it is necessary to evaluate the potential distribution and habitat suitability of these pest species. The damages caused by changes in the habitat of pests due to environmental changes require the use of technology that can predict potential habitats of pests in advance [18,19]. Species distribution modeling (SDM) is a method that predicts the potential habitat of a species and its variation owing to climate change, which is a function of climate and environmental variables. There are a few SDM algorithms; MaxEnt is one of the most popular SDM tools that use machine-learning methods to evaluate the possibility of occurrence using presence-only records [20]. This model implements maximum entropy theory to find a coordinate with similar model variable properties extracted from the occurrence coordinates of a target species. In addition, MaxEnt shows high model performance, even with a relatively small number of presence records [21,22]. Thus, current SDM studies have used it to evaluate potential risk areas of agricultural pests that mainly damage food crops such as rice and wheat [18,19,23,24,25].

Three *Leptocorisa* species are major pests of rice with a large economic impact, and the degree of habitat suitability, density, and habitat range is expected to increase because of climate change. Consequently, their damage to rice is expected to increase. Nevertheless, studies of SDM that predicts these species are limited, and none of the studies have simultaneously predicted and compared the potential distribution of species according to climate change. Therefore, in this study, the potential habitats of three *Leptocorisa* species in the present and future were mapped to analyze and predict current and potential risks according to climate change in Asia and Oceania. These results in this study can be used for pest risk assessment and management of the three *Leptocorisa* species.

## 2. Materials and Methods

### 2.1. Distribution Data Acquisition

The current distribution of the target species was obtained from a public species distribution database, such as the Centre for Agriculture and Bioscience International (CABI) and the Global Biodiversity Information Facility (GBIF), and previous studies that reported occurrence records. To minimize the uncertainty in the occurrence records, we compared the records from the three sources and confirmed the final occurrence points to be used in the modeling. We also acquired rice cultivation data and overlapped them after building the *Leptocorisa* species distribution models [26].

### 2.2. Leptocorisa chinensis Distribution

For *L**. chinensis*, CABI only provided its distribution in Japan [27], but GBIF showed more distributions in South Korea and southern and eastern China [28]. The distribution records in China seem to be reliable because there is a report that *L. chinensis* is more dominant in northern China because of its higher cold resistance than *L. acuta* [15]. It also has been reported that *L. chinensis* is present in South Korea [29,30,31]. In South Korea, a nation-scale field survey found that *L. chinensis* was mainly distributed in the southern regions, and few records were observed in central areas [31,32]. However, we decided to exclude a few central records in South Korea because the records might be temporary discoveries, owing to flight, and may not be enough to cause crop damage [33]. In addition, *L. chinensis* occurrence was confirmed in Southeast Asia (Bonin Island, Formosa, Thailand, etc.) [17,34,35], but it was impossible to obtain the specific coordinates; thus, we used them to test the developed model. Consequently, we used distribution records from South Korea, Japan, and China to develop a model for *L. chinensis*.

### 2.3. Leptocorisa acuta Distribution

*Leptocorisa acuta* shows relatively wide distributions among the species of the genus *Leptocorisa* [27,28], covering southern China, India, Southeast Asia, and Australia [35]. Both CABI and GBIF showed consistent distribution [27,28], except in the USA, and we removed them because no further study confirmed the record in the USA. We also used the occurrence data recorded in the previous study to build the model [36]. The records in Southeast Asian countries (Myanmar, Vietnam, Thailand, etc.) and Oceanian countries (Australia, Fiji, New Caledonia, etc.) did not provide specific coordinates for the development of the model; thus, they were used to test the model [15,37].

### 2.4. Leptocorisa oratoria Distribution

*Leptocorisa oratoria* is distributed from Southeast Asia to northern Australia [28]. Most distributions in both CABI and GBIF were consistent [27,28]. Despite the fact that there was a record of *L. oratoria* in Korea in CABI, we excluded it because the genus *Leptocorisa* was recorded as the species *L. chinensis* [38], and no previous study reported *L. oratoria* in South Korea. While *L. chinensis* was found mostly in temperate regions of Northeast Asia, *L. oratoria* is considered the most common species in tropical regions and a major rice pest in Southeast Asia [15]. To supplement the insufficient number of occurrence records, additional occurrence coordinates were included from previous studies [6,12,39,40,41,42,43]. Some previous studies have reported *L. oratoria* as a dominant species in tropical regions (Malaysia, Sumatra, Sri Lanka, and the Philippines) [6,8,39,44]. However, detailed coordinates could not be obtained; thus, we used the distributions in those regions to test the model.

### 2.5. Final Distribution Data

We could initially confirm the total of 150, 87, and 57 occurrence points for *L. chinensis*, *L. acuta*, and *L. oratoria*, respectively. To remove the redundant records and sampling bias caused by uneven sampling density, spatial filtering was applied with a 25 km radius with a consideration of climate heterogeneity in the occurrence points using the SDM toolbox in ArcGIS version 10.4.1 (ESRI, USA), resulting in 51, 59, and 53 points to be used to develop the model for *L. chinensis*, *L. acuta*, and *L. oratoria*, respectively [45,46] (Figure 1).

### 2.6. Model Variables Selection and Operation

Nineteen historical bioclimatic variables (1970–2000) and altitude data were obtained from WorldClim (www.worldclim.org) (accessed on 15 March 2021) with a resolution of 30 s of longitude and latitude [47]. Because decades are considered to be a short evolutionary period to adapt to a new climate and it is necessary to minimize the noise in climatic data, historical bioclimatic data were used to simulate the current potential distribution of the pests. In addition, the period of occurrence records includes the time span of historical bioclimatic variables, suggesting the bioclimatic variables could capture the characteristics of the occurrence areas. In addition to the above variables, land-cover data were obtained from the International Steering Committee for Global Mapping and converted into a suitable format to be used as an additional variable in the MaxEnt model [48]. After extracting the environmental variables at the occurrence coordinates of the three *Leptocorisa* species, Pearson’s correlation coefficient was calculated using IBM SPSS Statistics (version 21, IBM Statistics, Armonk, NY, USA). When selecting variables, correlation coefficients larger than ±0.7 and the variable contribution based on the default MaxEnt model with 21 environmental variables were considered to select the best environmental variables for model development [49] (Appendix A). Thereafter, bioclimatic variables related to high temperature, low humidity, and high precipitation were considered because they were reported to affect the three *Leptocorisa* species populations [15]. Therefore, nine environmental variables (Bio3, 5, 6, 8, 12, 17, 18, elevation, and land cover) were selected for *L. chinensis*, and ten (Bio2, 3, 5, 6, 8, 17, 18, 19, elevation, and land cover) and eight (Bio6, 7, 13, 17, 18, 19, elevation, and land cover) were selected for *L. acuta* and *L. oratoria*, respectively (Table 1, Appendix A).

In addition to spatial filtering performed to minimize sampling bias, we generated a bias background file using the kernel density function in R software [50,51,52] to weigh the occurrence records that could adjust uneven sampling density owing to the relatively small number of occurrence points [53]. Thereafter, we determined the regularization multiplier (RM) and feature combination (FC) under the selected environmental variables using the ENMeval package [54]. The optimal FC for *L. chinensis* and *L. acuta* were linear-quadratic (LQ) features; however, it was linear-quadratic-hindge (LQH) for *L. oratoria*. The optimal RM for *L. chinensis*, *L. acuta*, and *L. oratoria* were 2, 0.5, and 4, respectively. After determining the optimal conditions for the MaxEnt operation, we executed three *Leptocorisa* species models with 10-fold cross-validation [20].

### 2.7. Climate Change Scenario

To apply the climate change scenario, the Shared Socioeconomic Pathway (SSP) 245 for years 2081–2100, generated by the MIROC-6 model, was obtained from WorldClim (www.worldclim.org) (accessed on 15 March 2021) with a resolution of 30 s [47,55]. We used SSP245 to build the models by assuming a moderate emission scenario to avoid extreme simulation, and it predicted that the temperature would increase up to 4.2 °C, CO₂ emission at 550 ppm, and the precipitation also will be varied [56,57]. We selected 2081–2100 for a suitable period, not too short or too long to observe the change in future pest distribution.

### 2.8. Model Performance Test

We evaluated the performance of the model using three metrics that are generally used for the presence-only model: area under the receiver operating characteristic curve (AUC), true skill statistic (TSS), and omission rate 10% (OR10%). In general, an AUC of 0.5 means that model prediction is not better than random, values of >0.7 signify reasonable model prediction, and values of >0.9 indicate high performance [57,58,59]. TSS is a more practical and realistic metric for model performance compared with the AUC; it evaluates a model based on the selected threshold [60]. In this study, we used the maximum training sensitivity plus specificity logistic threshold for the TSS threshold because it has a robust value independent of prevalence [61]. In general, TSS values of >0.8 are considered to be excellent; however, good (TSS = 0.6–0.8), fair (TSS = 0.4–0.6), poor (TSS = 0.2–0.4), and no predictive ability (TSS < 0.2) have been reported [62,63]. OR10% is set as the threshold that excludes 10 percent of the localities with the lowest predicted values. This value of 0.1 is expected for an ideal model [64].

## 3. Results

### 3.1. The Results of Model Performance Test for the Three Leptocorisa Species

In this study, the AUC, TSS, and OR10% were calculated for the model performance test of *L. chinensis*, *L. acuta*, and *L. oratoria* (Table 2).

### 3.2. Potential Distribution of Three Leptocorisa Species in Asia and Oceania

Among the environmental variables used for *L. chinensis*, the greatest contribution was assigned to the minimum temperature of the coldest month, land cover, isothermality, and the precipitation of the warmest quarter. The model projected a high occurrence possibility of *L. chinensis* in warm–temperate zones, such as the southern parts of Korea, Japan, and central China. In China, Anhui province has high occurrence possibility values, and in Japan, the southern areas, where pecky rice damage has been reported, have high values [15]. After climate change, it is expected to expand to the eastern area of the Korean Peninsula, the Liaodong Peninsula in China, and Hokkaido in Japan (Figure 2, Appendix A).

The potential distribution map of *L. acuta* showed high values mainly in subequatorial and subtropical regions, including Southeast Asia and Oceania, corroborating previous records [15,37]. In this model, the precipitation of the warmest quarter had the highest contribution, followed by the minimum temperature of the coldest month, land cover, elevation, and isothermality, whereas mean diurnal range, the precipitation of the driest quarter, the maximum temperature of the warmest month, the mean temperature of the wettest quarter, and the precipitation of the coldest quarter contributed little. As a result of applying the SSP245 climate change scenario, *L. acuta*, which has the widest distribution among the *Leptocorisa* species, was predicted to have a similar distribution range to the current climate (Figure 3, Appendix A).

For *L. oratoria*, the precipitation of the wettest month and the minimum temperature of the coldest month showed high contributions. Other environmental variables, the precipitation of the driest quarter, the annual temperature range, land cover, the precipitation of the warmest quarter, the precipitation of the coldest quarter, and the elevation contributed little. In the potential distribution map of *L. oratoria*, the possibility of occurrence was high, mainly in tropical areas, corroborating previous reports [6,8,39,44]. Based on climate change, it was predicted that a higher density would be observed in most tropical areas (Figure 4, Appendix A).

Additionally, we overlaid the potential distribution maps of the three *Leptocorisa* species with a habitat threshold determined based on maximum training sensitivity plus specificity logistic thresholds of the three species [65,66]. After climate change, the potential distribution of *L. chinensis* expanded to the north of Japan, China, and the Korean Peninsula. No significant difference was found in *L. acuta* and *L. oratoria* after climate change. In the current climate, the possibility of the three *Leptocorisa* species distribution is in a wide latitudinal range in Asia and Oceania. The three species in the south coast area of China overlapped in the model, whereas two species (*L. acuta and L. oratoria*) overlapped in the subequatorial and subtropical regions of Southeast Asia (Thailand, Vietnam, Cambodia, etc.), the near-coast area of the western region of India, the northern region of Australia, and the southern region of the Melanesian Archipelago overlapped. Most of the potential distributions of the three *Leptocorisa* species coincided with rice cultivation areas except for the Australian regions (Figure 5). In particular, the high rice production areas (more than 10,000,000 tones production), and the potential distribution areas of *Leptocorisa* species overlapped in the middle area of China and the Java islands of Malaysia (Figure 6), suggesting a high potential risk of damage by these pests and the necessity for an intensive control of them.

## 4. Discussion

In this study, we developed MaxEnt models for three *Leptocorisa* species and projected the potential distribution according to climate change. The model performance metrics of AUC, TSS, and OR10% showed excellent performance for predicting the target species, according to the standards for the metrics mentioned, suggested that the developed models were sufficient to explain the possibility of the occurrence of the species. The performances of the models were high in this order: *L. chinensis*, *L. acuta*, and *L. oratoria*. In *L. oratoria*, compared to a fairly long latitude range, many of the occurrence points that were used to build the model are from Java Island, which might have influenced the model performance matrix [67]. In addition, the relatively small number of environmental variables used for the species could have affected the evaluation of the model [68]. The potential distribution of *L. acuta* covered the widest latitudinal range among the three *Leptocorisa* species. In contrast, the projected distribution of *L. chinensis* was the smallest in terms of latitudinal range, covering southern China to northern Japan. This might be because the occurrence records for warm–temperate zones were dense, causing relatively similar characteristics to be trained by the model. Consequently, the model performance metrics were the highest for *L. chinensis*, while *L. oratoria* had the lowest performance metrics. In addition, OR10% and TSS showed more prominent differences than AUC. This indicates that TSS or OR10% could be more effective than AUC, even though TSS can vary depending on the prevalence of the species [60,69,70]. In terms of model performance evaluation, the current metrics, i.e., AUC and TSS, should be carefully used due to their dependency on prevalence, even though the TSS is considered to be practical with high-quality occurrence records [60,69,71,72]. For this reason, it is essential to generate reliable distribution information and accurate background files that spatially consider the sampling bias. Without high-quality presence-absence records, which are practically common in species distribution modeling, another metric, similarity/F-measures, which is not biased by prevalence, could be a better option [70].

*Echinochloa crus-galli* and *E. colonai* are hosts of *L. oratoria*, which are widely distributed in Asia and Oceania. In addition, field observations in Korea revealed that *L. chinensis* was commonly found in *Digitaria ciliaris* and *Setaria viridis* communities, which have a wide distribution in Asia and Oceania [28,73,74]. Because the maximum flight distance of *L. chinensis* was estimated to be 24.2~29.4 km in a day, it could quickly disperse to areas that are under suitable conditions to find a host [33]. These facts imply that host plant distribution is unlikely to be a range-limiting barrier for *Leptocorisa* species due to its high mobility, emphasizing the importance of the climate for the distributions.

In general, the minimum temperature of the coldest month influenced the model prediction and largely contributed to the building of the three *Leptocorisa* species models. This seems to be affected by the ecological characteristics of the three *Leptocorisa* species, which are distributed in tropical, subtropical, and temperate zones of the warm region and have limitations in the northern and southern areas due to their own thermal developmental thresholds [15,29,75,76]. In particular, *L. oratoria*, which is the most sensitive to cold among these three species, showed the highest minimum temperature of the coldest month percent contribution (38.5%) to the model compared to the models of other species [15]. The precipitation of the warmest quarter or the wettest month also showed a significant effect on the models, which is consistent with a previous study that showed the significant impact of precipitation on the *Leptocorisa* species population [15]. The warmest quarter is generally the active period of *Leptocorisa* species, and humidity could affect their mortality. For example, the humidity was found to affect the egg hatching of *L. oratoria* [11,77]. In addition, if there is rainfall over a period of months, it would generate a greater supply of food for *Leptocorisa* species, and they could establish a larger population [15]. The land cover showed the second highest contribution in the *L. chinensis* model and the third highest contribution in the *L.*
*acuta* model. Specifically, the distribution possibilities were high in urban areas and needleleaf evergreen forests for *L. chinensis*, urban areas and paddy fields for *L.*
*acuta*, and paddy fields and broadleaf evergreen forests for *L.*
*oratoria*. *Leptocorisa* species are able to fly to areas where a host plant is prevalent; thus, the importance of land cover as a model variable suggests the role of the host plants in addition to the climatic conditions [28,33,73,74]. In addition, it is reported that *L. chinensis* hibernates in needleleaf evergreen forests [29,78], while the strong correlation between broadleaf evergreen forests and precipitation may be related to *L.*
*oratoria* distribution, as shown in the model contribution [79]. Therefore, it is possible to narrow the area where pests can be distributed through the land cover that represents the environmental characteristics of the area. Isothermality was a relatively high environmental factor and showed the third highest contribution in the *L. chinensis* model. If isothermality is too high, the seasonal temperature difference cannot be enough to induce hibernation, which is a positive factor for the reproduction of *L. chinensis* [80].

In contrast to the acquisition of georeferenced records of *L. chinensis* in East Asia, it was not possible to obtain specific coordinates in Southeast Asia where *L. chinensis* distribution was recorded [17,34,35]. Hence, the potential distribution map cannot show the occurrence potential in Southeast Asia. Nevertheless, *L. chinensis* is not a dominant pest compared to *L. oratoria* or *L. acuta*, suggesting that the prediction is at least useful for risk assessment of *L. chinensis* damage in East Asia, such as Japan, Korea, and China. The model of *L. acuta* occurrence possibility shows a high occurrence possibility in New Caledonia, Fiji, the northern area of Vietnam, Bangladesh, and Assam and Meghalaya, India, where *L. acuta* was recorded [15,37,81,82,83]. *L. oratoria* has been recorded as a dominant pest among *Leptocorisa* species in the Philippines, Malaysia, Sri Lanka, and Sumatra [6,8,39,44] and is consistent with the model outcome that predicted the highest occurrence possibility of this pest in these areas among the three species. The model in this study represents the high occurrence probability of *L. oratoria* in Borneo, which is economically significant in tidal swamp rice agroecosystems [15,84]. Moreover, the model indicates that greater pecky rice damage can occur after climate change in Cambodia, and Vietnam, around the Mekong River area, and on Java island, where rice farming is active [85].

According to the records, *L. acuta* and *L. oratoria* are distributed together in Bhutan, India, China, Taiwan, Japan (Okinawa), the Philippines, Myanmar, Thailand, Vietnam, Malaysia, Indonesia, and Australia [35]. This is consistent with the results predicted in this study. In addition, the southern region of China and Taiwan are places where the three species overlap in the model. In these areas, environmental factors other than climate could more influence the interspecies competition of the species than in the other areas; therefore, confirming the species that dominates can help in performing detailed ecological studies of these species.

*L**eptocorisa**chinensis* has been regarded as a rice pest in Japan, causing pecky rice [29,86,87]. Although the economic damage caused by pecky rice has not been reported in Korea, if *L. chinensis* expands northward and increases its abundance owing to climate change, it may become a rice pest, as in Japan [29]. The main rice yields in Korea in the individual provinces are in this order: Jeollanam-do, Chungcheongnam-do, Jeollabuk-do, Gyeongsangbuk-do, and Gyeonggi-do [88]. After climate change, most of the provinces will have a higher occurrence possibility of *L. chinensis* in 2081–2100. In particular, the potential distribution map shows a high probability of settlement in the Gyeonggi area in 2081-2100, where rice fields are abundant. The main rice yields in Japan in the individual provinces are in this order: Niigata, Hokkaido, Akita, Yamagata, and Ibaraki, and the possibility of occurrence in all of these provinces except for Niigata is expected to increase according to climate change [89]. In China, Hunan, Jiangxi, Heilongjiang, Anhui, and Jiangsu are the main rice cultivation areas [90]. In the case of Anhui province, the possibility of *L. chinensis* occurrence was relatively high.

The maps showed that the distribution range of *L. acuta* and *L.*
*oratoria* will not be dramatically changed by 2081-2100. However, the potential distribution in 2081–2100 suggests more damage to rice cultivation because *L. oratoria* density is expected to be higher than at present, and it is the dominant rice pest in Southeast Asia already [15]. Especially in Bangladesh and the Red River Delta region of Vietnam, as they are places where both rice production and the occurrence possibility of *L. acuta* and *L.*
*oratoria* are high.

As climate change progresses, in addition to expanding the distribution range of the *Leptocorisa* species, it is expected that the life cycle will be accelerated, the density will increase, and the rice ear development period will be earlier, which can increase the possibility of heavy damage to rice. Even though a few studies have been conducted on the preparations for mitigating the economic damage of *Leptocorisa* species, such as insecticides including natural extracts, entomopathogenic fungus, natural enemies and parasitic species, density survey, and resistant rice varieties, a study on their distribution ranges according to climate change is demanded [80,91,92,93,94,95,96,97,98].

In this study, we evaluated potential distributions of three *Leptocorisa* species and comprehensively identified the areas exposed to their damage with climatic, host, and environmental variables. Based on our analysis, there are some potential areas with suitable environmental condition for the pests, suggesting that cautious monitoring to prevent their invasion is necessary, and this study provides fundamental data to establish a strategy for it.

## Figures and Tables

**Figure 1 insects-13-00750-f001:**
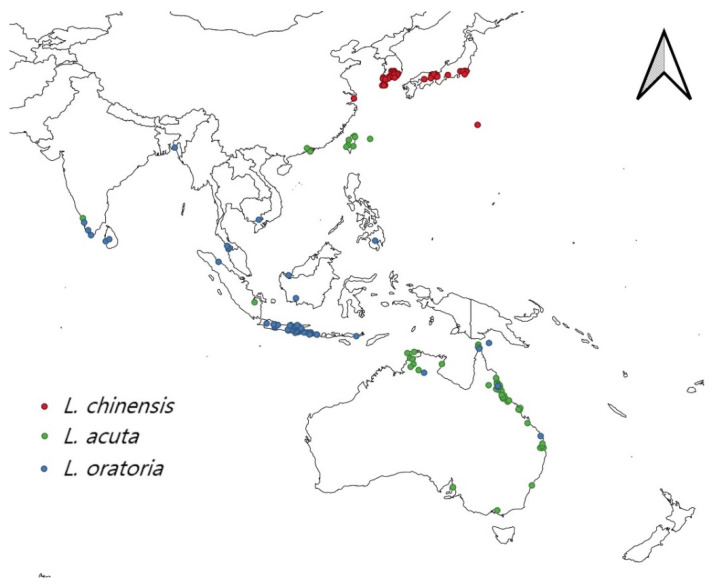
Occurrence records for three *Leptocorisa* species.

**Figure 2 insects-13-00750-f002:**
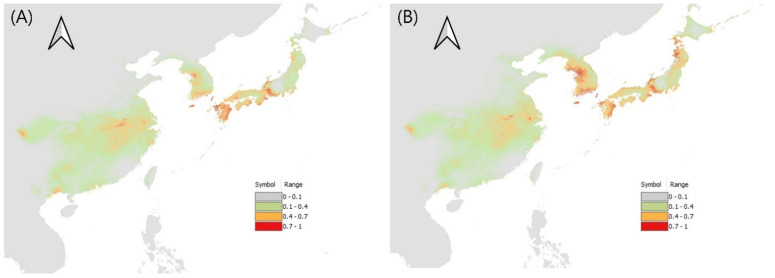
Potential distribution of *L. chinensis* (**A**) under current climate, and (**B**) for 2081–2100.

**Figure 3 insects-13-00750-f003:**
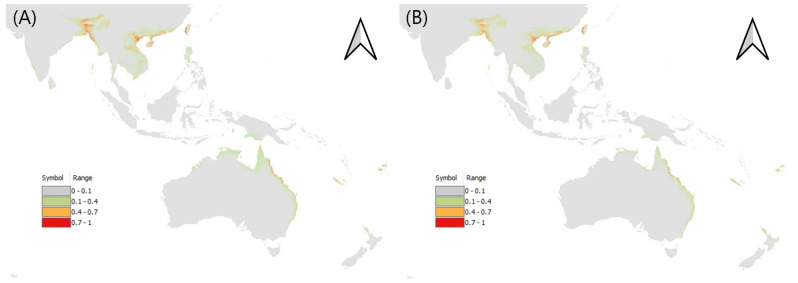
Potential distribution of *L. acuta* (**A**) under current climate, and (**B**) for 2081–2100.

**Figure 4 insects-13-00750-f004:**
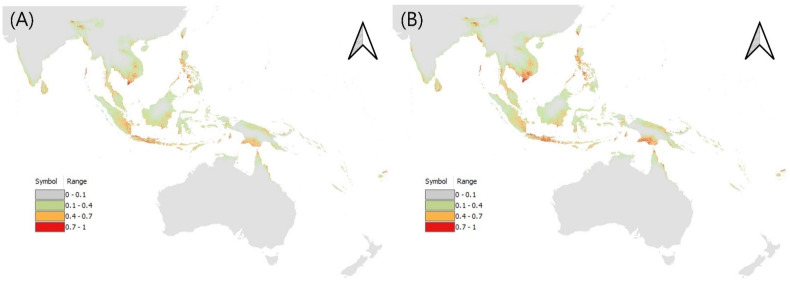
Potential distribution of *L. oratoria* (**A**) under current climate, and (**B**) for 2081–2100.

**Figure 5 insects-13-00750-f005:**
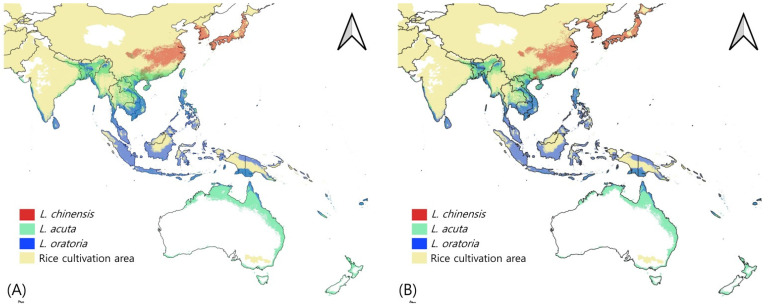
Integrative potential distribution of three *Leptocorisa* species with rice cultivation area (**A**) under current climate, and (**B**) for 2081–2100.

**Figure 6 insects-13-00750-f006:**
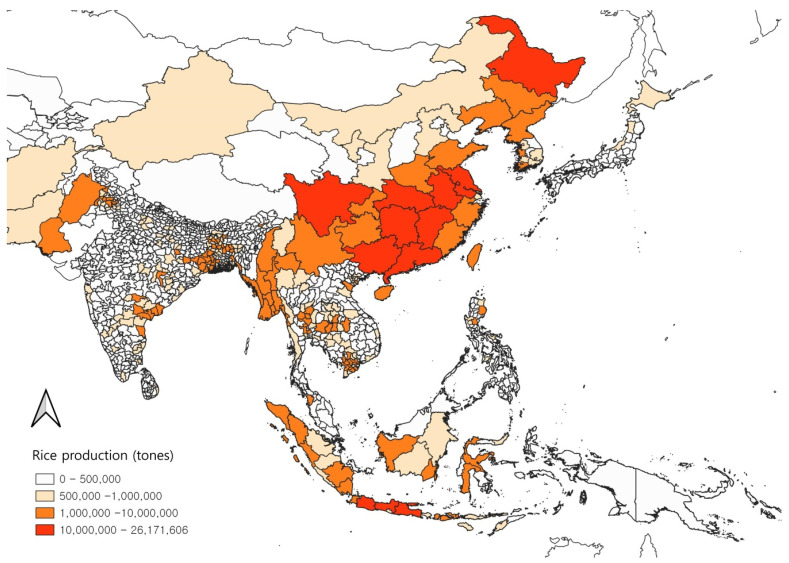
Map of rice cultivation areas with the amount of productions.

**Table 1 insects-13-00750-t001:** List of selected environmental variables for each species.

Variable Code ^a^	Description	Percentage Contribution		
		*L. chinensis*	*L. acuta*	*L. oratoria*
Bio2	Mean diurnal range ^b^	-	1.9	-
Bio3	Isothermality ^c^	18.8	4.8	-
Bio5	Maximum temperature of the warmest month	0	1.7	-
Bio6	Minimum temperature of the coldest month	30.1	24.8	38.5
Bio7	Temperature annual range (Bio5–Bio6)	-	-	6.5
Bio8	Mean temperature of the wettest quarter	7.9	0.2	-
Bio12	Annual precipitation	5	-	-
Bio13	Precipitation of wettest month	-	-	39.9
Bio17	Precipitation of the driest quarter	0	1.8	7
Bio18	Precipitation of the warmest quarter	10.6	44.5	1.2
Bio19	Precipitation of the coldest quarter	-	0.2	0.6
Elevation	Altitude data	3.3	6.9	0.4
Land cover	Land covers with 20 classifications	24.3	13.2	6.0

^a^ Variable codes were obtained from WorldClim (www.worldclim.org). ^b^ Mean diurnal range = mean of monthly (maximum temperature − minimum temperature). ^c^ Isothermality = mean diurnal range/temperature annual range × 100.

**Table 2 insects-13-00750-t002:** Measures of the models for three *Leptocorisa* species.

Measure	*L. chinensis*	*L. acuta*	*L. oratoria*
Test AUC	0.993	0.980	0.980
TSS	0.958	0.915	0.905
OR10%	0.160	0.170	0.187

## Data Availability

Data is contained within the article or Appendix A.

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
