# Peer review of "Evaluation of Spatial Distribution of Three Major Leptocorisa (Hemiptera: Alydidae) Pests Using MaxEnt Model"

_insects, 2022, doi:10.3390/insects13080750_

Round 1

Reviewer 1 Report

This study deals with a recently popular approach in predicting potential distribution of a species and includes some valuable point of view in using SDMs for predicting suitable areas for three Leptocorisa pests of rice crops in Asia and Oceania. This manuscript is very well written and prepared. As far as the extraction of environmental data go, this investigation was very well designed, replicated, and executed. The results are remarkably consistent by all measurements taken. The analytical approach is appropriate, and the data are clearly presented. The results are nicely placed within the context of the current, relevant literature and the conclusions are justified based on the results. This is very well prepared, and I am unable to offer anything of consequence to further improve this very nice manuscript. This should receive quite a bit of interest from the readership of this journal and those interested in SDMs and Leptocorisa incursion management.  

Author Response

Thank you for your comments. All the answers and revisions were indicated in the attachment for all reviewers. 

Reviewer 2 Report

The study shows a potential distribution of three pest bugs based on previously published and new distribution records.

Only a few points need to be amended:

Line 42: This first occurrence of the three scientific names should be completed by names of authors of description. The last name must be written Leptocorisa oratoria (Fabricius, 1764) (syn. L. oratorius (Fabricius, 1764)). 

Line 99. Thailand, Australia.

Table 1: right bottom value: 6.0

Author Response

Thank you for your comments. All the answers and revisions were indicated in the attachment for all reviewers. 

This manuscript is a resubmission of an earlier submission. The following is a list of the peer review reports and author responses from that submission.

Round 1

Reviewer 1 Report

It seems that climatic models resulting in potential distribution maps of three bugs were constructed properly. But the selection of only well-documented distribution data prior to the modelling might make the model too narrow. I do not understand what means if some localities were not used for model construction but were used for testing. Were in Results are these tests?

Suggestions for minor changes: 

l.30: words present in the abstract need not be repeated in keywords;

l.34-35: If the list is not complete, is it necessary at all?

l.41: Add author names in this first appearance of the three names.

Use the feminine form L. oratoria throughout.

l.94: In the figure, there are a lot of points in north Australia. Also India. - I do not understand the difference between the map and the text. 

l.119: Coordinates are numbers. Do you display sample sites?

l.122: What does it mean? 10 min of longitude and latitude?

l.126: Correct would be the forward selection of variables by the software without preselection by authors.

l. 127: P-value as a selection criterion is expected.

Table 1: Polarity of the variables must be indicated. E.g. "Precipitation of the warmest quarter" - is required lower than a specific threshold or higher?

Table 2: It is usually not accepted to double such information in text and table. Unless you can say something more about the meaning of the values. 

l.175 and many others: Using these codes is useless for readers. They will appreciate the full names of the variables. 

l.176 and others: These percentages are already in the table. 

l.190 and others: word "habitat" - You perhaps mean range of climatic parameters. Habitat is rather the type of vegetation, which was not mentioned in the species characteristics above. 

l.228: Dependence of the model on the number of presence records means that the initial omission of some data causes errors in the modelling.

l.229: "Coordinates" - you mean sites. Coordinates are always two. 

l.256-257: No such relationships are obvious. Explain why isothermality is related to hibernation. 

l.263 and others: Unclear what you are writing about. Area or yield in individual provinces?

l.322-324: Repeated from the previous paragraph.

Reviewer 2 Report

The work concerns the modeling of the distribution of various organisms, which has been very popular in recent years. The study uses selected climatic data from the last 30 years of the twentieth century for the places of occurrence of species of the genus Leptocorisa, and on this basis, the occurrence of the species is predicted in the years 2081-2100.

My basic reservations are primarily caused by the very sense of conducting and publishing such analyzes, in a situation of so many unknowns and reservations, I will try to list them.

  1. Some of the assumptions are not verifiable, eg that the present occurrence of a particular species is conditioned by the climatic factors included in the study. To find this out, the squares of the UTM grid or other map projection should be drawn and the occurrence of the species should be determined, and then correlated with the climatic data.
  2. It is not known whether the same factors will influence the occurrence of the species in several dozen years as they do now.
  3. The analysis took into account the point places where the species was found, and not the dynamics of its abundance. These results were compared with the climatic factors averaged for thirty years. If the 30-year observations of the insect occurrence were presented in comparison with the climatic data, the prediction would certainly be better.
  4. Selected scenarios take into account changes in temperature and CO2, while the factors determining the presence of both the tested insects and the host plant - rice are rainfall. This leads to absurd results such as mapping the potential ranges of rice-feeding insects in the Australian desert.
  5. For me it is incomprehensible why modeling was chosen in 60-80 years and not, for example, in 20 or 100 years?
  6. The authors do not state the power of their tests.

My criticism concerns not only this work, but the entire research stream, which I consider to be of little use to modern science. Therefore, I believe that the work should be rejected.

Author Response

Thank you for your criticism. I'm sorry you think that way and understand your concerns on the model-based approach. We tried our best to answer your concerns on the study.  Please, see the attachment.

Reviewer 3 Report

This article by Jeong Ho Hwang and colleagues is well motivated, the structure is appropriate, and the manuscript is well written without missing any key details. The methods used are appropriate for the objectives of the work and, in general, well depicted.

The resulting figures and maps are sufficient, informative, and of good quality helping to follow the reasoning throughout the manuscript. The discussion of results and comments on future research was nicely done and will be useful to others. Overall, I enjoyed reading the manuscript. A few remarks have been made below for authors to consider:

Lns152-165: The authors have clearly explained thresholds behind the AUC and TSS metrics. What are the potential improvements of TSS upon AUC (Allouche et al. 2006; Lobo et al. 2008; Jiménez-Valverde 2012; 2014)? Please note that both metrics, TSS and AUC, are now considered outdated (see Leroy et al. 2018). Presence-only metrics (without pseudo-absences in the contingency evaluation table are now more preferred than TSS and AUC. It’s okay if you keep them but be aware of all these issues, and the potential pitfalls of TSS and AUC metrics should be discussed in more details. 

Lobo JM, Jiménez-Valverde A, Real R (2008) AUC: a misleading measure of the performance of predictive distribution models. Glob Ecol Biogeogr 17:145–151. https://doi.org/10.1111/j.1466-8238.2007.00358.x

Jiménez-Valverde A (2012) Insights into the area under the receiver operating characteristic curve (AUC) as a discrimination measure in species distribution modelling. Glob Ecol Biogeogr 21:498–507. https://doi.org/10.1111/j.1466- 8238.2011.00683.x

Jiménez-Valverde A (2014) Threshold-dependence as a desirable attribute for discrimination assessment: Implications for the evaluation of species distribution models. Biodivers Conserv 23:369–385. https://doi.org/10.1007/s10531-013-0606- 1

Allouche O, Tsoar A, Kadmon R (2006) Assessing the accuracy of species distribution models: Prevalence, Kappa and the True Skill Statistic (TSS). J Appl Ecol 43:1223–1232

Leroy B, Delsol R, Hugueny B, et al (2018) Without quality presence–absence data, discrimination metrics such as TSS can be misleading measures of model performance. J Biogeogr 45:1994–2002. https://doi.org/10.1111/jbi.13402 

Good luck!

Author Response

Your comment gives me great motivation. We tried to answer for your comment, and revised manuscript as best as we can. Please, see the attachment.

Round 2

Reviewer 2 Report

Many thanks to the authors for their explanations. Unfortunately, they do not convince me to accept working with a clear conscience. I just think that these kinds of publications are a waste of science, Sorry. I have been discussing this for a long time with my colleagues who use these mathematical techniques, and they cannot convince me to accept this approach. Perhaps our grandchildren will find out who is right today in eighty years. And for me, the conclusion is that I should not consent to the review of such works, and the decision to accept this work must be made by the editor.

Author Response

Thank you for your opinion. I understand your criticism on this type of work, and one of my career task is to persuade someone who doesn't like research like this. I always fully respect others opinion. I think it's not a matter of right or wrong, but a difference in approach.